# High Strength Construction Material Based on Sulfur Binder Obtained by Physical Modification

**Sergey Sergeevich Dobrosmyslov** [1,2,*], **Vladimir Efimovich Zadov** [1], **Rashit Anvarovich Nazirov** [2], **Gennady Efimovich Nagibin** [2], **Anton Sergeevich Voronin** [1,2], **Mikhail Maksimovich Simunin** [1,2], **Yuri Vladimirovich Fadeev** [1,2] and **Stanislav Viktorovich Khartov** [1]

[1] Federal Research Center Krasnoyarsk Scientific Center, Siberian Branch,
Russian Academy of Sciences (FRC KSC SB RAS), 660036 Krasnoyarsk, Russia; zeus1948@yandex.ru (V.E.Z.);
a.voronin1988@mail.ru (A.S.V.); michanel@mail.ru (M.M.S.); daf.hf@list.ru (Y.V.F.); stas_f1@list.ru (S.V.K.)

[2] Department of Building Design and Real Estate Expertise, Faculty of Industrial and Civil Engineering,
School of Engineering and Construction, Siberian Federal University, 660041 Krasnoyarsk, Russia;
nazirovra@gmail.com (R.A.N.); nagibin1@gmail.com (G.E.N.)

\* Correspondence: dobrosmislov.s.s@gmail.com

**Abstract:** In this work, a method for obtaining a high-strength composite material on a sulfur binder without the use of chemical modifiers was proposed. It consists in obtaining a thixotropic casting mixture in the vicinity of the yield point of the system during vibration laying. The compressive strength of the obtained composite was about 97.5–94.0 MPa. Physical and mechanical characteristics remained stable for 7 years. The samples were obtained for a model composition of sulfur/marshalite (finely ground 98% silicon dioxide). The microstructure of the synthesized material was studied by electron microscopy, the results of which showed that shrinkage cavities are characteristic of a material with a low density, and no shrinkage cavities were found for a high-strength material. The phase composition was determined by the methods of XRD analysis, according to the results of which it can be argued that sulfur is in the orthorhombic form ($S_8$). This technique can be proposed for obtaining the high-strength stable building material.

**Keywords:** sulfur concrete; sulfur; silicon oxide; physical and mechanical properties; shrinkage; physical modification





## 1. Introduction

The growing volumes of the world production of sulfur create preconditions for the search for new areas of its use. Almost all sulfur (more than 90%) is produced today as a by-product of oil and gas processing and non-ferrous metallurgy when utilizing sulfur dioxide [1,2].

Currently, oil, gas and metallurgical companies are actively developing projects focused on the processing of sulfur into new products, the sale of which is more focused on the construction and road industries [3–6]. Despite the large-scale consumption in these areas (asphalt-concrete mixtures, sulfur binder, sulfur concrete), sulfur has not yet found such widespread use in them as in traditional consumption segments-fertilizer production [7]. One of the promising areas for the use of sulfur concrete is the construction of buildings in the absence of water, for example, on Mars [8,9].

The molecular structure of sulfur has a very wide variety of polymorphic modifications. Currently, more than thirty sulfur allotropes have been identified, most of them are insufficiently studied, and there is still no unified classification for them [10–12].

The solidified melt of pure sulfur does not have sufficient strength due to a significant change in the specific volume [13] (about 13%) during the transition from the liquid phase to the monoclinic, then to the rhombic phase. In this regard, the basis of the technology for obtaining high-quality and durable compositions based on sulfur binder is the modification

of sulfur. This is the transformation of the crystal structure into plastic, stabilization of the polymer modification of sulfur, which improves the physical and mechanical characteristics and chemical resistance of sulfur compositions [14–20]. In the manufacture of a product, sulfur is melted and modified with mineral powders and chemical additives [21–23], obtaining a more durable sulfuric binder. The essence of chemical modification is the use of additives to stabilize the molecular chain during partial polymerization [24]. In this case, the modifier prevents the formation of crystalline sulfur or directly destroys the crystalline structure. As a result, shrinkage processes decrease and slow down, and the strength of products increases [23]. A decrease in shrinkage processes leads to a decrease in internal stresses, and their slowdown leads to a gradual relaxation. These factors ultimately increase the strength of the material. However, it should be noted that the strength does not exceed 65–70 MPa [25–30], which fully meets the requirements for building products. One of the first proposed chemical modifiers is dicyclopentadiene [31]; Figure 1 shows the chemical modification by dicyclopentadiene.

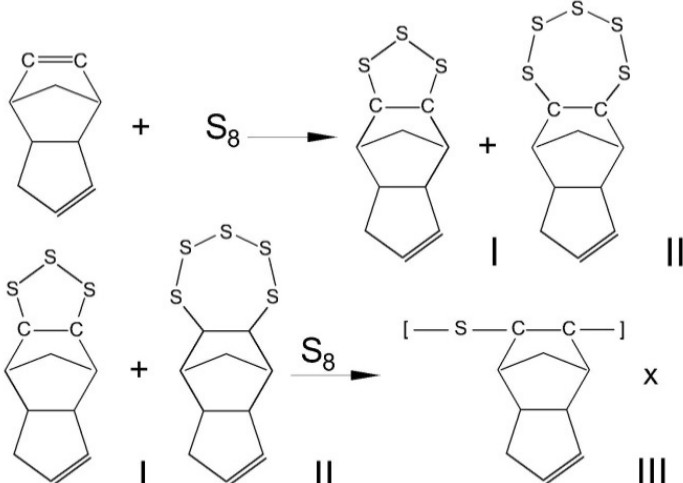

**Figure 1.** Chemical modification with dicyclopentadiene (2-column fitting image).

The use of various modifiers, primarily complex organic compounds, allows the stabilization of polymer sulfur, which leads to an increase in the physical and mechanical characteristics of the finished product. In this case, sulfur partially enters into a chemical reaction with the modifier, forming complex molecules. At the same time, the main disadvantage of chemical modification is the high cost of organic modifiers, which complicates their use in construction. Stabilization is temporary, the strength of the modified material decreases over time. As a disadvantage of chemical modification, the toxicity of modifiers can be noted [31], which complicates their use in the composition of sulfur compositions.

The paper proposes a fundamentally new approach to solving the problem of hardening of sulfur concretes, using the main disadvantage of the sulfur matrix-significant shrinkage during the phase transition from a liquid to a solid state. The strength of the composite material is limited by the weakest structures; for conventional sulfur concretes, it is a matrix consisting of sulfur, which has volumetric defects due to significant volumetric shrinkage during solidification. The proposed approach consists in obtaining a high-strength composition based on the creation of a composite with a minimum spreading ratio, which is about 13 percent by volume of the binder volume, i.e., corresponds to an increase in the volume of the binder (sulfur matrix) during its melting.

Obviously, the minimum spreading ratio can be achieved by using the maximum possible amount of mineral additive when filling the entire interpore space with liquid sulfur, provided that the filler components are wetted with sulfur. At the same time, the maximum packing density for balls of the same size is about 74% [32]. In reality, the filler particles have a shape other than spherical and different sizes. At the same time, the packing

density differs from ideal by several percent. Accordingly, the optimal concentration of sulfuric binder is in the range of 20–40%. At the same time, it can be noted that this statement is true under the condition of complete wetting of the filler surface with liquid sulfur; with the complex shape of filler particles, this result can be obtained by mechanical action, which can be carried out through intensive mixing.

When the sulfur binder cools and solidifies, shrinkage phenomena are localized in a small area of the shrinkage cell, in which the particles of the high-strength mineral filler come into mechanical contact, which, when loaded, leads to a significant transfer of stresses from a relatively weak matrix to a high-strength filler and, in general, the effect of a relatively weak sulfur matrix on physical and mechanical characteristics is reduced. This increase in physical and mechanical characteristics is a physical modification, that is, the formation of a strong skeleton from a mineral component with the localization of shrinkage processes in a small area.

Accordingly, it can be noted that to obtain a high-strength composite material on a sulfur binder, it is necessary to use a high-strength and highly dispersed powder with a minimum particle size (with high adhesion to the sulfur matrix) adhesively interacting with sulfur. At the same time, with the propagation of cracks in a brittle material, their deceleration occurs on various inclusions and inhomogeneities [33], therefore, a high concentration of the mineral additive leads to an increase in physical and mechanical characteristics.

## 2. Materials and Equipments

The minimum spreading ratio was achieved by increasing the amount of the mineral component. For a high-quality alignment of the filler and matrix, significant mechanical forces are required for the sulfur-mineral filler system, which were achieved due to a high stirring speed (200 rpm).

Samples were made as follows. Industrial sulfur (98% purity, a specific weight of 1.032 g/cm$^3$) was melted with constant vigorous stirring at a temperature of 150 °C. Then the mineral components preheated to 150 °C were added. The experimental setup is shown in Figure 2.

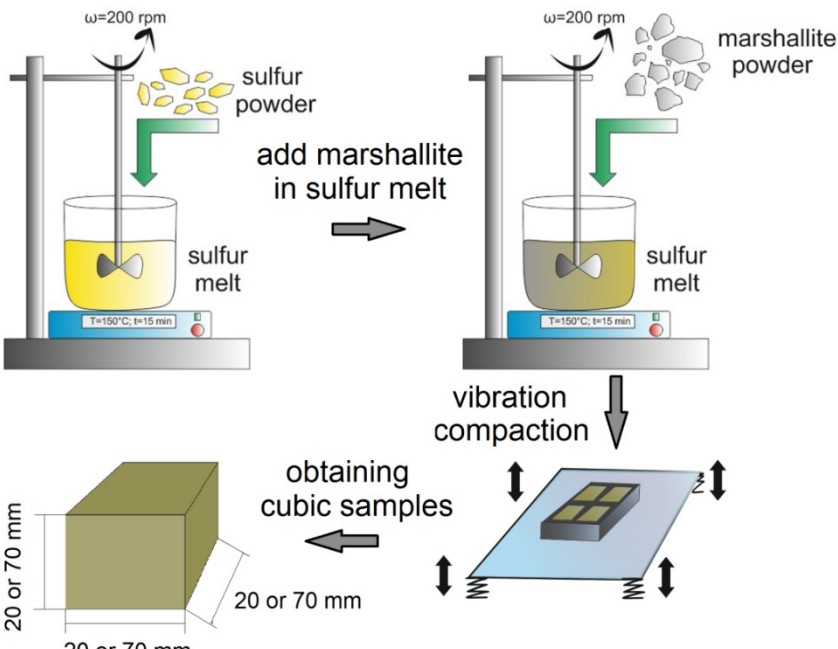

**Figure 2.** Diagram of an experimental setup for the manufacture of material on sulfur binder (sulfur concrete) (single-column fitting image).

In order to prevent the formation of significant internal stresses in the material during the phase transition from liquid to solid, the mixture was overheated to a temperature of 150–160 °C. A further increase in temperature led to an increase in the viscosity of the system (polymeric sulfur is formed). In addition, the mold was heated up to a temperature of 150–160 °C.

Evaluation studies were carried out on samples of 20 × 20 × 20 mm, due to their simpler manufacture. The results obtained were verified on samples with dimensions of 70 × 70 × 70 mm taken for testing building materials. Of course, in our case, only 20 × 20 × 20 mm samples could be used, since the characteristic sizes of the particles used are many times smaller than the sizes of the samples. However, it is standard for building materials to use larger specimens. Therefore, all measurements were repeated.

As component soft he composition were used: Marshalite (Technical silicon dioxide 98%). As part of the work in the manufacture of samples, the amount of sulfur binder varied from 30 to 70%. Marshalite was used to obtain a model line-up. To test the compressive strength and density, specimens with dimensions of 20 × 20 × 20 mm and 70 × 70 × 70 mm were made.

Diffraction data were obtained on a PANalyticalX'Pert PRO diffractometer with a PIXcel detector. X-ray diffraction patterns were taken with CuK$\alpha$ radiation with a graphite monochromator in the 2θ range from 5° to 90° with a step of 0.013°. Full-profile phase analysis was carried out using the derivative difference minimization (DDM) method [34].

The morphological study of the composites was carried out by scanning electron microscopy on a Hitachi TM 4000 microscope anufacturer "Hitachi High-Technologies Corporation", address 24-14, Nishi-Shimbashi 1-chome, Minato-ku, Tokyo, 105-8717, Japan, equipped with a powder diffractometer D8 Advance X-ray microanalysis attachment; address Germany, Bruker AXS GmbH, Ostliche Rheinbrückenstr. 49, 76187 Karlsruhe, Germany.

*Mechanical Characterization*

The compressive and flexural strength was studied according to the methodology presented in the C109/C109M13 Standard Test Method for Compressive Strength of Hydraulic Cement Mortars. Density was determined according to ASTM C 642–13, Standard Test Method for Density, Absorption, and Voids in Hardened Concrete, ASTM International, West Conshohocken.

Test press Instron 600DX-B1-C3-G7B (60 tons). Manufacturer information: Instron, Purpose: Testing of building materials. Scope: Determination of strength characteristics of materials. Characteristics: Maximum load 60 tons, 825 University Avenue Norwood, MA 02062-2643 USA.

Instron 3360 Series Testing Press (5 tons). Manufacturer information: Instron, USA. Purpose: Testing of building materials. Scope: Determination of strength characteristics of materials. Characteristics: Maximum load 5 tons, 825 University Avenue Norwood, MA 02062-2643 USA.

Compressive and flexural strengths were tested according to the method described in C109/C109M 13 Standard Test Method for Compressive Strength of Hydraulic Cement Mortars. Density was determined according to ASTM C 642–13, Standard Test Method for Density, Absorption, and Voids in Hardened Concrete, ASTM International, West Conshohocken. An Instron 3369 testing machine was used as a press.

### 3. Results and Discussion

The Figure 3 shows the results of granulometric analysis.

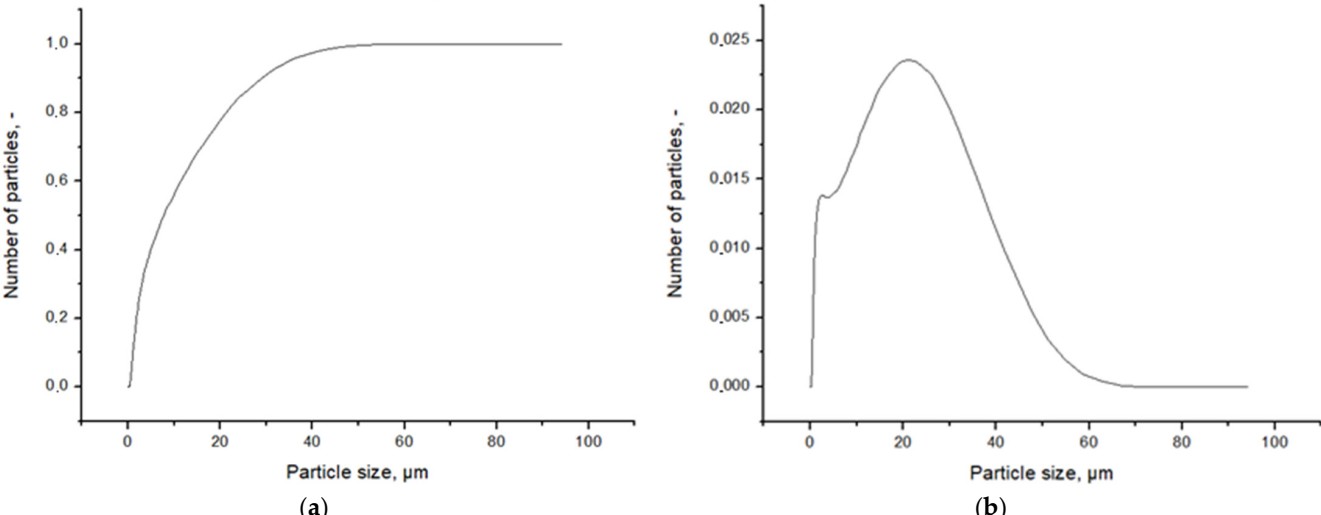

**Figure 3.** Results of granulometric analysis, (**a**) integral curve, (**b**) differential curve.

To study the phase composition of sulfur after the preparation of the composite, XRD analysis was performed. Figure 3 shows an X-ray diffraction pattern of a sample of 70% marshalite−30% sulfur.

It follows from the results obtained that the sample contains 70% quartz and 30% sulfur S8 with a characteristic orthorhombic crystal lattice belonging to the space group Fddd. The results allow us to conclude that the proportion of sulfur in the material does not change. An increase in the low-angle intensity indicates the presence of amorphous components in the material, which should be attributed to polymer sulfur.

To assess the shrinkage processes in the structure of sulfur concrete, micrographs of sulfur/marshalite compositions (50/50 and 35/65) were obtained. Composition of the mass. 50% sulfur-mass. marshalite 50% for visual demonstration of cavities, and composition of mass 35% sulfur-mass marshalite 65% to demonstrate their absence. Figure 4 shows the results of electron microscopy of samples of the sulfur/marshalite composition (50/50), the circles indicate morphological defects formed as a result of sulfur shrinkage during its solidification.

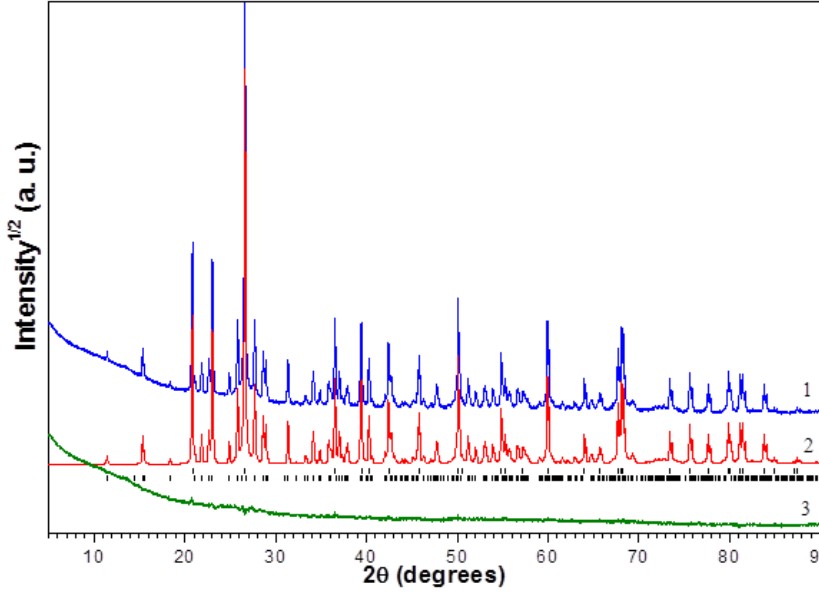

**Figure 4.** X-ray diffraction pattern of a sample of 70% marshalite−30% sulfur, curves: experimental (1), calculated (2) and difference (3). (2-column fitting image).

As follows from Figure 5, the size of the cavity is comparable to the size of the mineral component. Such defects can affect the strength of the material. In general, sulfur completely covers the particles of silicon oxide, but at the same time, the sulfur content of about 50% leads to the formation of shrinkage cavities of significant size, which softens the material. Elemental mapping by energy dispersive X-ray spectroscopy illustrates the separation of the sulfur and marshalite phases. At ×500 magnification, it can be seen that sulfur densely wets marshalite particles and forms a homogeneous interface.

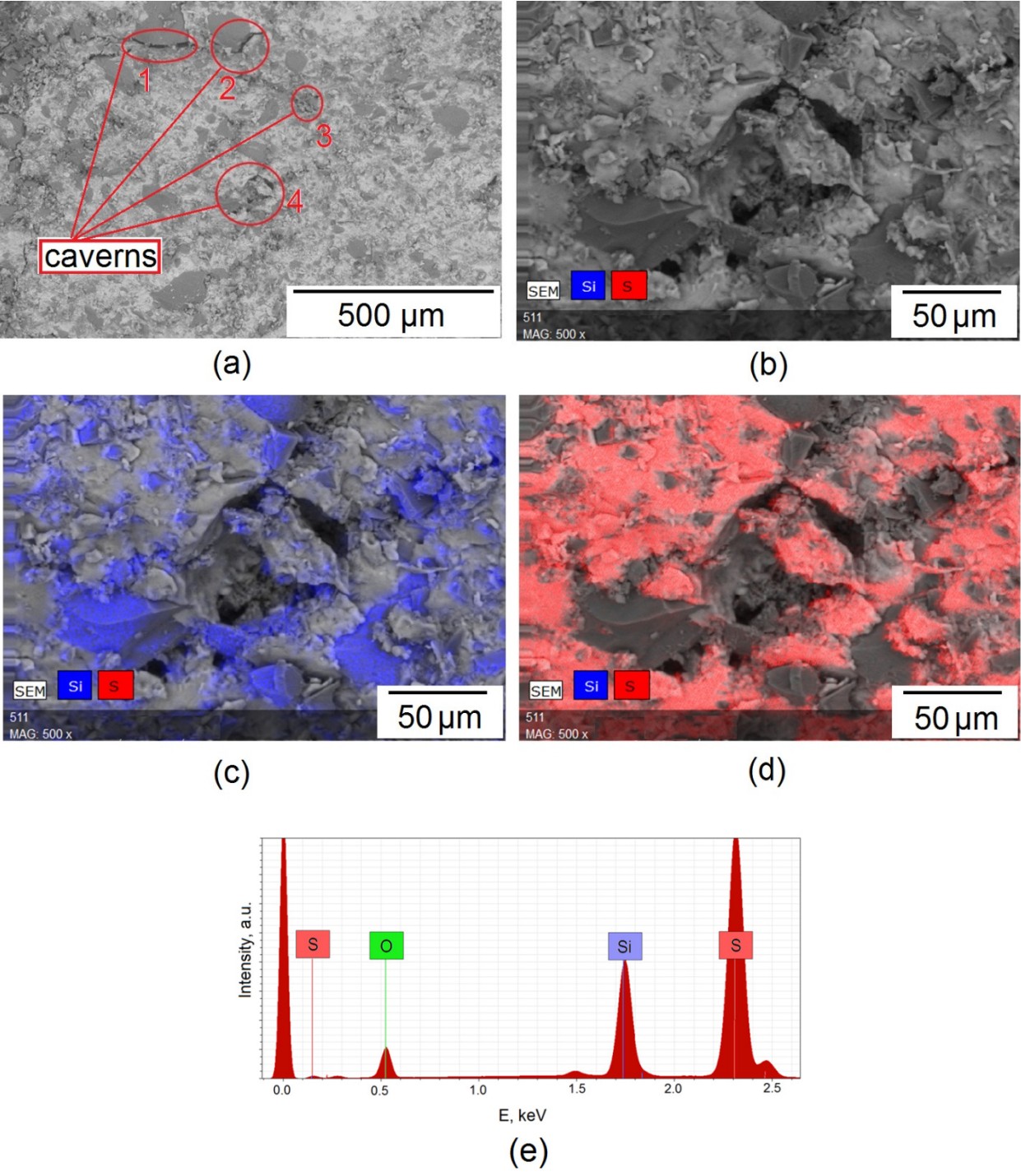

**Figure 5.** Micrographs of the sulfur/marshalite composite (50/50), in the figure (**a**) caverns formed as a result of shrinkage are marked, (**b**) the cavern is approximated, (**c**) the distribution of silicon, (**d**) the distribution of sulfur, the results of (**e**) X-ray luminescence analysis. (2-column fitting image).

The morphology of the 35%/65% sulfur-marshalite composition is characterized by dense packing of the filler in sulfur (Figure 6). As follows from the presented results, the concentration of sulfur binder of about 35% does not lead to the appearance of shrinkage cavities and allows you to create a continuous dense media. In this case, sulfur sticks together silicon oxide particles, which is indicated by the presence of a dense interface between them during elemental mapping by energy dispersive X-ray spectroscopy. In general, the resulting material has low porosity. The results of electron microscopy (Figure 6) show that the sulfur is uniformly distributed and acts as a binder.

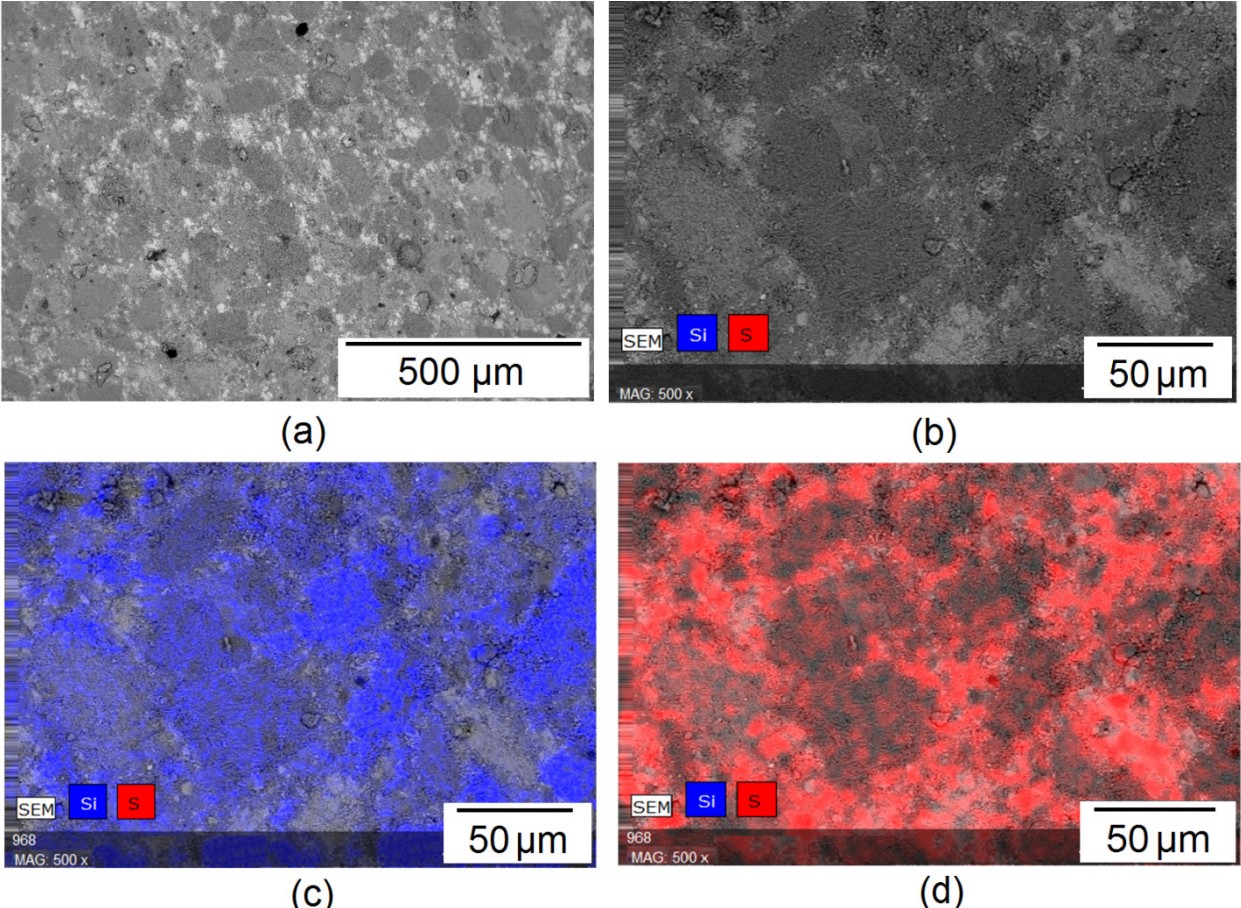

**Figure 6.** Micrographs of a sulfur/marshalite composite (35%/65%).(**a**) microstructure, (**b**) microstructure c approximate, (**c**) distribution of silicon, (**d**) distribution of sulfur. (2-column fitting image).

The main goal was to determine the uniformity of distribution of the sulfur binder, as well as to evaluate the interaction of the sulfur binder with the mineral filler (wetted or not at the micro level).

Physicomechanical tests of the obtained samples confirm the presented results of electron microscopy. Figure 7a shows typical fracture curves of the material 65% marshalite−35% sulfur, for 7 samples of the same composition, made in 2014 and 7 years later, without the use of chemical modifiers.

Figure 7b,c shows the values of the density and compressive strength of the material on a sulfur binder and a photograph of a sample obtained in 2014. The threshold of fluidity during vibration compaction for these materials is about 30–35% of the mass amount of sulfur, therefore, in the vicinity of this concentration, the step was reduced.

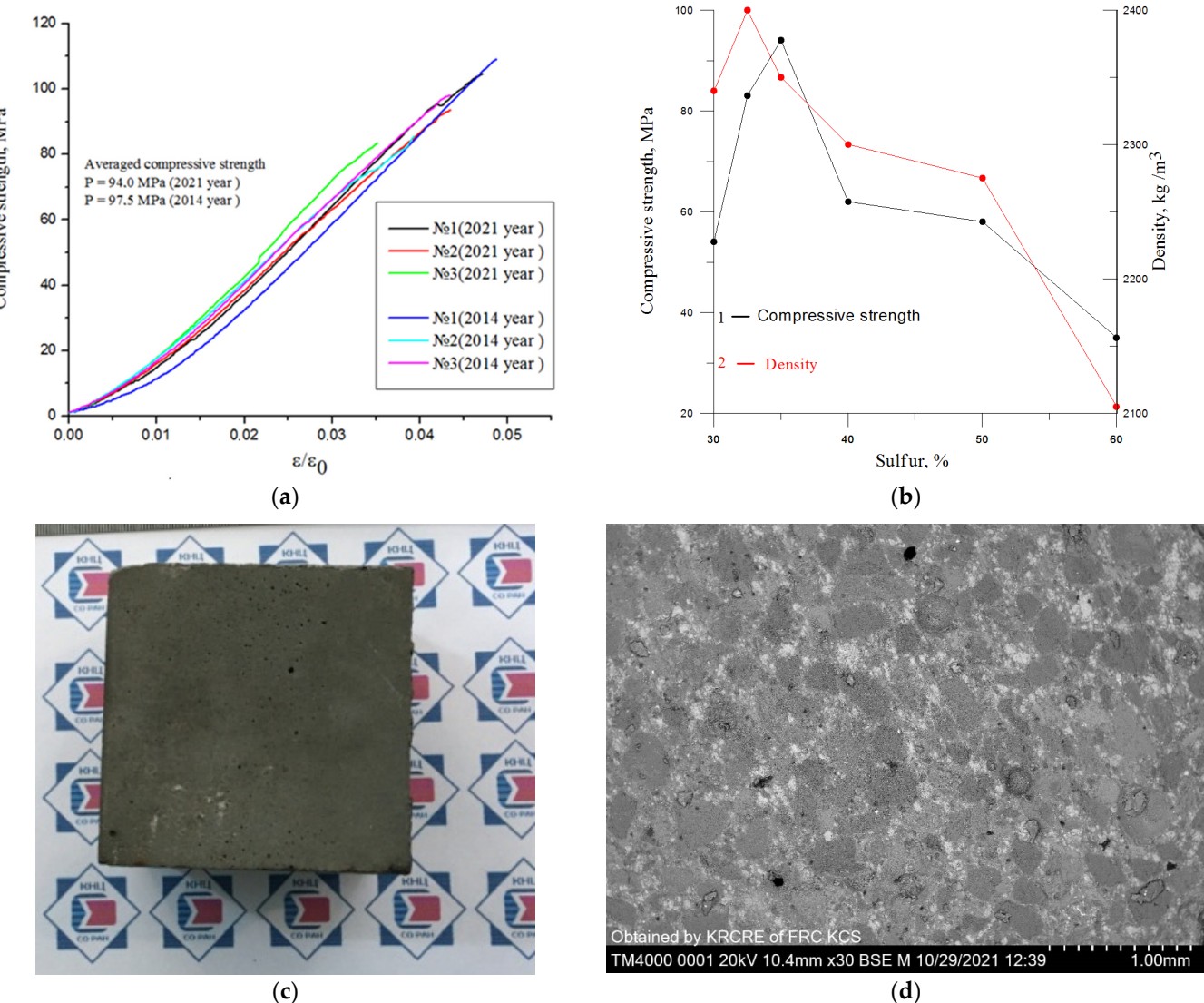

**Figure 7.** Compressive strength test results for a sulfur/marshalite composite (35%/65) (**a**), dependence of strength and density on sulfur concentration (**b**), photographs of a sample obtained in 2014 (2-column fitting image), sample obtained in 2014 and tested in 2021 (**c**), samples after testing (**d**).

Due to the fact that sulfur has a large number of allotropic compounds, the essence of chemical modification is to stabilize the polymer component. However, at the same time, orthorhombic sulfur ($S_8$) remains thermodynamically stable at room temperature, and modifiers only slow down the process of transition from polymeric to orthorhombic. Within 6–24 months, most of the polymeric sulfur turns into rhombic. The exception is the material in the production, the amount of modifier used (10–15%) is comparable to sulfur, while the cost of production of building products increases significantly.

According to the above graph, with a decrease in sulfur concentration starting from 60%, an increase in density and strength occurs (Figure 6b), and an increase in density and strength is observed simultaneously. This result is related to two factors. First, the density of mineral components is higher than the density of sulfur. Second, the packing density increases, and the effect of shrinkage processes decreases. The maximum strength for a composite using marshalite is ave97.5 MPa (Figure 6a), this strength for a composite based on sulfur binder without the use of chemical modifiers was achieved for the first time [7,17,23,29,35,36]. Comparison of the results obtained with the literature data is presented in Figure 8.

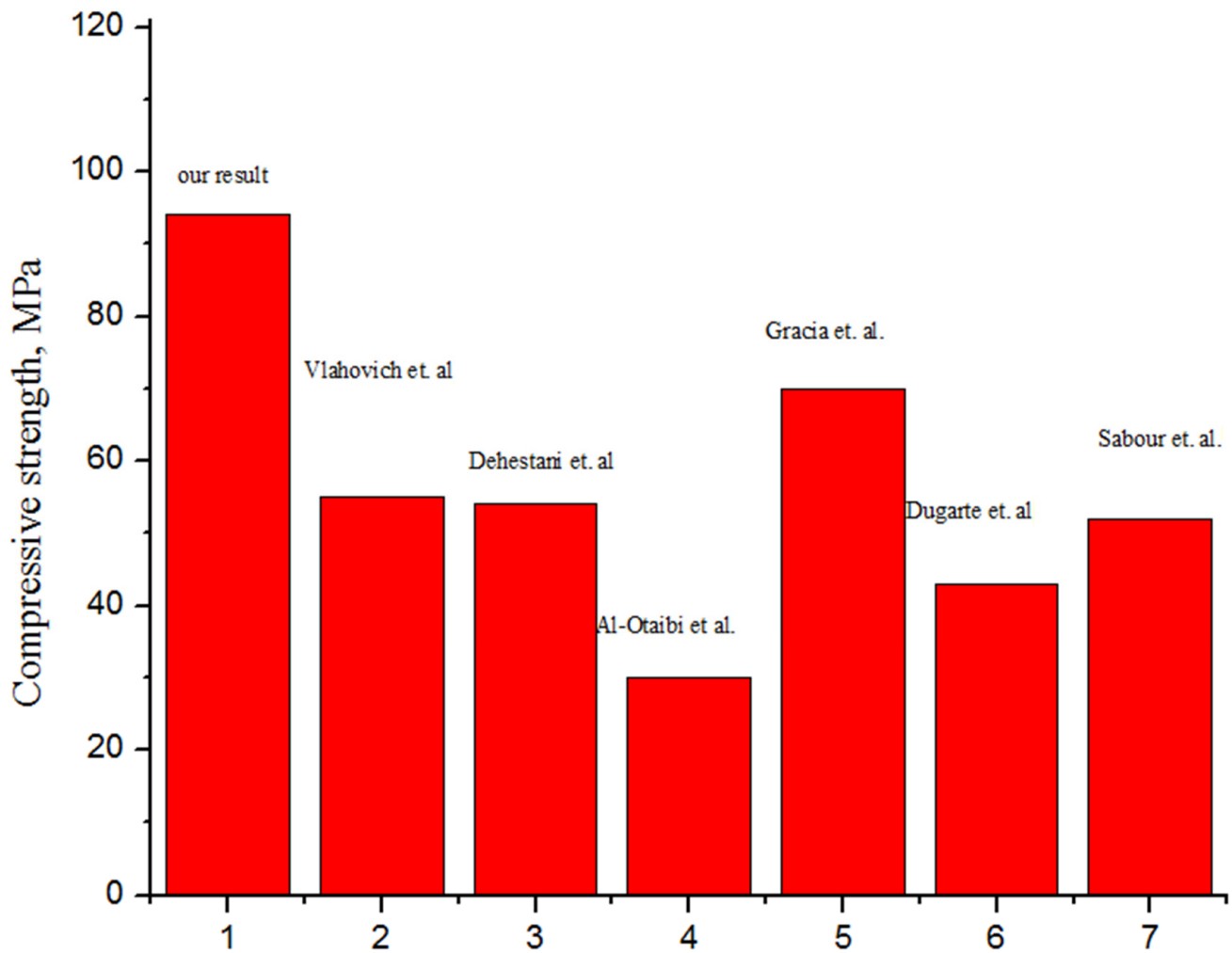

**Figure 8.** Comparison of the obtained results and literature data (single-column fitting image) [5,19,25,31–33].

Based on the presented results; it can be concluded that sulfur concrete is a classic composite; and the applied strengthening methods (physical modification) are more promising than chemical modification. The optimal concentration of the sulfur binder is about 35% of the mass of the finished composite; thus, forming a dense and virtually non-porous material

Marshallite particles can be classified as spherical, therefore, the fraction of void volume between filler particles with their dense packing cannot be lower than 26% [32]. Knowing the density of quartz (~2.61 g/cm$^3$) and the density of orthorhombic sulfur (~2.07 g/cm$^3$), we indicate that the theoretical mass fraction of sulfur should be above 22%. However, starting from values less than 35% sulfur, the density and strength of the material begins to fall, this is due to a change in the rheological properties of the system, and as a result, a deterioration in workability.

A study of the morphology of the composite material indicates a high density of the composite material 65% marshalite−35% sulfur. This result shows that there were no phase transitions in the structure of the sulfur binder after crystallization, and the sulfur itself is in a crystalline form, this fact is confirmed by the results of XRD analysis (Figure 4). Thus, it can be concluded that the high strength of sulfur concrete is ensured by a combination of factors-a homogeneous filler, the plasticity of sulfur during laying, and the formation of a homogeneous fine-grained phase of orthorhombic sulfur.

The first samples of a high-strength composite, without the use of chemical modifiers, on a sulfur binder were obtained in 2014. To assess the stability of the physical and mechanical characteristics, the sample was kept for 7 years. The uniformity of the properties

of the samples indicates that all the processes of the formation of orthorhombic sulfur took place during the packing of the samples. The results presented in Figure 6b, were re-obtained in 2021. Figure 6c shows a photograph of a sulfur concrete cube obtained in 2014, which did not collapse under a load of about 92 MPa.

### 4. Conclusions

This paper proposes a fundamentally new approach to solving the problem of hardening sulfur concrete, localizing shrinkage processes in small volumes uniformly distributed over the product.

This result is achieved by using: a minimum concentration of a sulfur binder (while maintaining the percolation of the system), a finely dispersed strong aggregate (the size of the voids between close-packed aggregate particles is comparable to the average particle size) and intensive mixing in the vicinity of the yield threshold with subsequent vibration compaction of the suspension. At the same time, a high concentration of filler leads to inhibition of cracks during fracture, which also increases the physical and mechanical characteristics. This technique was proposed for the first time for sulfur concrete. Stabilization of the physical and mechanical characteristics of sulfur concrete is achieved by localizing shrinkage caverns in extremely small volumes of the shrinkage cell (physical modification), which implies the use of a high-strength and highly dispersed filler with a particle size of about 50 microns or less.

Physical modification makes it possible to obtain a high-strength material, compressive strength 97.5 MPa, without the use of chemical modifiers, while the physical and mechanical characteristics remain stable 7 years after synthesis. At the same time, the physical modification of sulfur concretes has a number of advantages over the chemical, primarily due to the fact that in the structure of the composite all physicochemical processes end within a few weeks after the formation of the material. The method of physical modification can be considered as a promising direction for obtaining high-strength products with stable physical and mechanical characteristics.

**Author Contributions:** Conceptualization, S.S.D. and V.E.Z.; methodology, A.S.V., R.A.N. and M.M.S.; software, Y.V.F.; validation, R.A.N. and G.E.N.; formal analysis, V.E.Z.; investigation, S.S.D., Y.V.F., A.S.V. and V.E.Z.; resources, G.E.N.; data curation, R.A.N. and S.V.K.; writing—original draft preparation, S.S.D.; writing—review and editing, S.S.D. and V.E.Z.; visualization, A.S.V. and Y.V.F.; supervision, S.V.K.; project administration, S.V.K.; funding acquisition, S.V.K. All authors have read and agreed to the published version of the manuscript.

**Funding:** This research received no external funding.

**Institutional Review Board Statement:** Not applicable.

**Informed Consent Statement:** Not applicable.

**Data Availability Statement:** Not applicable.

**Acknowledgments:** Studies electron microscopy were performed on the equipment of Krasnoyarsk Regional Center of Research Equipment of Federal Research Center «Krasnoyarsk Science Center SB RAS». This research did not receive any specific grant from funding agencies in the public, commercial, or not-for-profit sectors.

**Conflicts of Interest:** The authors declare no conflict of interest.

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
