# Peer review of "High Strength Construction Material Based on Sulfur Binder Obtained by Physical Modification"

_buildings, doi:10.3390/buildings12071012_

Round 1
Reviewer 1 Report
Article:
HIGH STRENGTH CONSTRUCTION MATERIAL BASED ON SULFUR BINDER OBTAINED BY PHYSICAL MODIFICATION
In this work, a method for obtaining a high-strength composite material on a sulfur binder without the use of chemical modifiers was proposed.
The abstract could be revised with quantitative results.
Some more latest studies are required in the introduction to enhance the novelty of this study.
What is the symbol w in Figure 1?
Lines 119-125, what is this? which language the authors are using, the articles must be in English.
Section 2.1, need to provide more details about the different machines.
Figure 3, caption b is in a different language.
Ultimate failure modes of tested specimens are not presented in the form of figures.
Also, results are presented in a simple way, please provide scientific reasons behind them.
Conclusions are too long, it's better to be concise.
Author Response
Article:
HIGH STRENGTH CONSTRUCTION MATERIAL BASED ON SULFUR BINDER OBTAINED BY PHYSICAL MODIFICATION
In this work, a method for obtaining a high-strength composite material on a sulfur binder without the use of chemical modifiers was proposed.
The abstract could be revised with quantitative results.
We added to the abstract the results obtained after 7 years of exposure(94MPa).
Some more latest studies are required in the introduction to enhance the novelty of this study.
A sentence has been added to the introduction (lines 35-36):
One of the promising areas for the use of sulfur concrete is the construction of buildings in the absence of water, for example, on Mars [Obtaining elemental sulfur for Martian sulfur concrete / Aaron Barkatt and Masataka Okutsu// Journal of Chemical Research March-April 2022 1–17 https://doi.org/10.1177/17475198221080729, Research progress on lunar and Martian concrete / Zhuojun Hu a , Tao Shi a,b,* , Muqiu Cen a , Jianmin Wang c,* , Xingyu Zhao d , Chen Zeng a , Yang Zhou a , Yujian Fan a , Yanming Liu a , Zhifang Zhao// Construction and Building Materials 343 (2022) 128117 https://doi.org/10.1016/j.conbuildmat.2022.128117]. 1
The following references have also been added:
Analysis of Compressive Strength of Sulfur Concrete / Eri S. Romadhon and Achmad Hanif // Journal of Mechanical, Civil and Industrial Engineering https://DOI:10.32996/jmcie
You, X.M. (2021) Research Progress of the Modification in Sulfur Concrete. Materials Sciences and Applications, 12, 353-361. https://doi.org/10.4236/msa.2021.127024
What is the symbol w in Figure 1?
Perhaps you mean Figure 2? In this figure, w stands for the number of turns per minute.
Lines 119-125, what is this? Which language the authors are using, the articles must be in English.
Sorry, this paragraph has not been changed. Your note has been corrected.
Section 2.1, need to provide more details about the different machines.
It is not entirely clear what data is needed? Need a factory serial number? Two press machines were used, one up to 5 tons, the other up to 100 tons. The machine for 5 tons did not have enough power to crush cubes with dimensions of 70 * 70 * 70 mm.
Figure 3, caption b is in a different language.
This mistake has been corrected
Ultimate failure modes of tested specimens are not presented in the form of figures.
sample obtained in 2014 and tested in 2021(c), samples after testing(d)
Also, results are presented in a simple way, please provide scientific reasons behind them.
Changed the third paragraph from the end.
Marshallite particles can be classified as spherical, therefore, the fraction of void volume between filler particles with their dense packing cannot be lower than 26% [31]. Knowing the density of quartz (~2.61 g/cm3) and the density of orthorhombic sulfur (~2.07 g/cm3), we indicate that the theoretical mass fraction of sulfur should be above 22%. However, starting from values less than 35% sulfur, the density and strength of the material begins to fall, this is due to a change in the rheological properties of the system, and as a result, a deterioration in workability.
Changed the second paragraph from the end.
A study of the morphology of the composite material indicates a high density of the composite material 65% marshalite - 35% sulfur. This result shows that there were no phase transitions in the structure of the sulfur binder after crystallization, and the sulfur itself is in a crystalline form, this fact is confirmed by the results of XRD analysis (Fig. 4). Thus, it can be concluded that the high strength of sulfur concrete is ensured by a combination of factors - a homogeneous filler, the plasticity of sulfur during laying, and the formation of a homogeneous fine-grained phase of orthorhombic sulfur.
Conclusions are too long, it's better to be concise.
Removed the last paragraph in the conclusions. Shortened the last sentence of the second paragraph

Reviewer 2 Report
I have read the ms HIGH STRENGTH CONSTRUCTION MATERIAL BASED ON SULFUR BINDER OBTAINED BY PHYSICAL MODIFICATION. The ms could contribute to the literature concerning concrete’s manufacturing.
The goal is clear, to determine the uniformity of distribution of the sulfur binder, as well as to evaluate the interaction of the sulfur binder with the mineral filler (wetted or not at the micro level). However, the ms needs to be checked for language. Some of the lines are difficult to follow/understand. E.g. line 304 to 310 and there are lines that are off.
Please check line 119-125
Line 20 - orthoric form? Is this really correct?
Line 158 - The figure 3 shows the results of granulometric analysis. Delete ‘the’
Line 226 -227 - Previous studies show that chemical modification leads to a temporary increase in physical and mechanical properties. Add references!
Line 250 - orthormic sulfur? Please check
Author Response
Reviewer 2
I have read the ms HIGH STRENGTH CONSTRUCTION MATERIAL BASED ON SULFUR BINDER OBTAINED BY PHYSICAL MODIFICATION. The ms could contribute to the literature concerning concrete’s manufacturing.
The goal is clear, to determine the uniformity of distribution of the sulfur binder, as well as to evaluate the interaction of the sulfur binder with the mineral filler (wetted or not at the micro level). However, the ms needs to be checked for language. Some of the lines are difficult to follow/understand. E.g. line 304 to 310 and there are lines that are off.
Thanks, the paragraph at 304-310 has been removed.
Please check line 119-125
Sorry, this paragraph has not been changed. Note corrected.
Line 20 - orthoric form? Is this really correct?
This is a typo, we corrected it to "orthorhombic"
Line 158 - The figure 3 shows the results of granulometric analysis. Delete ‘the’
Corrected
Line 226 -227 - Previous studies show that chemical modification leads to a temporary increase in physical and mechanical properties. Add references!
This result is based on personal experience and the use of various chemical modifiers. Long-term studies (for several years) were not found in the literature, so the text was removed.
Line 250 - orthormic sulfur? Please check
We corrected it to "orthorhombic"

Reviewer 3 Report
1. This paper contains lots of missing figures. Thus reviewing process is not possible at this point. Even though certain figure numbers are described in the text, but no figures are found.
1) Fig. 7b is missing (Line 232)
2) 97.5 MPa (Fig. 6a) (Line 236): Which figure do authors mean?
3) The results are presented in Figure 6c (Line 279). But figure 6c represents photographs of a sample.
4) Figure 7d (lines 280) is missing.
2. In addition, Incomplete sentences are found in the following lines.
2-a) lines 143-146
2-b) Lines 147-149
2-c) Lines 174-176
3. One paragraph is written in Russian: Page 4, lines 119-125.
Author Response
- This paper contains lots of missing figures. Thus reviewing process is not possible at this point. Even though certain figure numbers are described in the text, but no figures are found.
1) Fig. 7b is missing (Line 232)
Sorry, this is a typo. Fixed on 6b
2) 97.5 MPa (Fig. 6a) (Line 236): Which figure do authors mean?
The measurement results are presented in this form for the purpose of demonstration and stability of physical and mechanical properties. In this figure, we wanted to demonstrate that the mechanics of the synthesized samples did not actually change over 7 years, so the failure curves were given. For easier perception, it is possible to add the obtained curves of single test results. If necessary, we will do so. The set of curves indicates the stability of the properties.
3) The results are presented in Figure 6c (Line 279). But figure 6c represents photographs of a sample.
Fixed on 6a
4) Figure 7d (lines 280) is missing.
Fixed on 6s
- In addition, Incomplete sentences are found in the following lines
2-a) lines 143-146
Test Method for Density, Absorption, and Voids in Hardened Concrete, ASTM International, West Conshohocken.
Test press Instron 600DX-B1-C3-G7B (60 tons). Manufacturer information: Instron, USA. Purpose: Testing of building materials. Scope: Determination of strength characteristics of materials. Characteristics: Maximum load 60 tons
2-b) Lines 147-149
Instron 3360 Series Testing Press (5 tons). Manufacturer information: Instron, USA. Purpose: Testing of building materials. Scope: Determination of strength characteristics of materials. Characteristics: Maximum load 5 tons.
2-c) Lines 174-176
To assess the shrinkage processes in the structure of sulfur concrete, micrographs of sulfur / marshalite compositions (50/50 and 35/65) were obtained. Composition of the mass. 50% sulfur-mass. marshalite 50% for visual demonstration of cavities, and composition of mass 35% sulfur-mass marshalite 65% to demonstrate their absence. Figure 4
- One paragraph is written in Russian: Page 4, lines 119-125.
Sorry, this paragraph has not been changed. Note corrected.

Round 2
Reviewer 2 Report
No reservation!